# Exogenous Albumin Is Crucial for Pig Sperm to Elicit In Vitro Capacitation Whereas Bicarbonate Only Modulates Its Efficiency

**DOI:** 10.3390/biology10111105

**Published:** 2021-10-26

**Authors:** Bruna Resende Chaves, Ana Paula Pinoti Pavaneli, Olga Blanco-Prieto, Elisabeth Pinart, Sergi Bonet, Márcio Gilberto Zangeronimo, Joan E. Rodríguez-Gil, Marc Yeste

**Affiliations:** 1Biotechnology of Animal and Human Reproduction (TechnoSperm), Institute of Food and Agricultural Technology, University of Girona, ES-17003 Girona, Spain; brunarufla@gmail.com (B.R.C.); anap.pavaneli@gmail.com (A.P.P.P.); elisabeth.pinart@udg.edu (E.P.); sergi.bonet@udg.edu (S.B.); 2Unit of Cell Biology, Department of Biology, Faculty of Sciences, University of Girona, ES-17003 Girona, Spain; 3Department of Veterinary Medicine, Federal University of Lavras, BR-37200-000 Lavras, MG, Brazil; zangeronimo@ufla.br; 4Laboratory of Andrology and Technology of Swine Embryos, Department of Animal Reproduction, School of Veterinary Medicine and Animal Science, University of São Paulo, BR-13635-900 Pirassununga, SP, Brazil; 5Unit of Animal Reproduction, Department of Animal Medicine and Surgery, Faculty of Veterinary Medicine, Autonomous University of Barcelona, ES-08193 Bellaterra (Cerdanyola del Vallès), Spain; obprieto@gmail.com

**Keywords:** sperm capacitation, pig, bicarbonate, bovine serum albumin (BSA), glycogen synthase kinase (GSK3), protein kinase A (PKA)

## Abstract

**Simple Summary:**

In this work, we addressed if the presence of exogenous bicarbonate required for pig sperm capacitation, which is a necessary step to acquire fertilizing ability. While sperm incubated in media without BSA or BSA/bicarbonate did not achieve in vitro capacitation, those incubated with BSA reached that status under any bicarbonate concentration, even when bicarbonate was absent. Interestingly, there were differences related to the concentration of bicarbonate, since sperm incubated in media with BSA and with no bicarbonate or 5 mM bicarbonate showed lower overall efficiency in achieving in vitro capacitation than those incubated in the presence of BSA and higher concentration of bicarbonate. Additionally, at the end of the experiment, sperm incubated in the presence of BSA and 38 mM bicarbonate showed lower motility and plasma membrane integrity than those incubated in media with BSA and lower concentrations of bicarbonate. In conclusion, BSA is crucial in for pig sperm to elicit in vitro capacitation and trigger the subsequent progesterone-induced acrosome exocytosis. In contrast, although exogenous bicarbonate does not appear to be indispensable, it shortens the time needed to reach that capacitated status.

**Abstract:**

This work sought to address whether the presence of exogenous bicarbonate is required for pig sperm to elicit in vitro capacitation and further progesterone-induced acrosome exocytosis. For this purpose, sperm were either incubated in a standard in vitro capacitation medium or a similar medium with different concentrations of bicarbonate (either 0 mM, 5 mM, 15 mM or 38 mM) and BSA (either 0 mg/mL or 5 mg/mL). The achievement of in vitro capacitation and progesterone-induced acrosomal exocytosis was tested through the analysis of sperm motility, plasma membrane integrity and lipid disorder, acrosome exocytosis, intracellular calcium levels, mitochondria membrane potential, O_2_ consumption rate and the activities of both glycogen synthase kinase 3 alpha (GSK3α) and protein kinase A (PKA). While sperm incubated in media without BSA or BSA/bicarbonate, they did not achieve in vitro capacitation; those incubated in media with BSA achieved the capacitated status under any bicarbonate concentration, even when bicarbonate was absent. Moreover, there were differences related to the concentration of bicarbonate, since sperm incubated in media with BSA and with no bicarbonate or 5 mM bicarbonate showed lower overall efficiency in achieving in vitro capacitation than those incubated in the presence of BSA and 15 mM or 38 mM bicarbonate. Additionally, at the end of the experiment, sperm incubated in the presence of BSA and 38 mM bicarbonate showed significantly (*p* < 0.05) lower values of motility and plasma membrane integrity than those incubated in media with BSA and lower concentrations of bicarbonate. In conclusion, BSA is instrumental for pig sperm to elicit in vitro capacitation and trigger the subsequent progesterone-induced acrosome exocytosis. Furthermore, while exogenous bicarbonate does not seem to be essential to launch sperm capacitation, it does modulate its efficiency.

## 1. Introduction

Mammalian sperm are unable to fertilize oocytes upon ejaculation because, despite being mature and motile, they need to reside within the female reproductive tract and interact with that environment [1]. Within the female reproductive tract, sperm undergo capacitation, which was first described in the 1950s [2,3]. Following this, sperm are able to interact with oocyte vestments, trigger acrosome exocytosis and subsequently fuse with the oocyte plasma membrane (reviewed in [4]). The capacitated spermatozoon undergoes a series of functional changes that include, amongst others, modifications in the composition of the sperm plasma membrane, acrosome remodeling, an increase in mitochondrial activity and sperm motility, a noticeable rise in intracellular Ca^2+^ and ROS levels, and the tyrosine-phosphorylation of certain sperm proteins [5,6,7,8,9,10].

At present, sperm capacitation can be performed in vitro in several (including pigs), although not all, mammalian species. Because effective in vitro sperm capacitation is required for successful in vitro fertilization (IVF), the main attempts to establish a proper capacitation medium have been related to the development of this assisted reproductive technique. Most previous studies have aimed to set a medium mimicking the oviductal environment, where the capacitation status is fully achieved (reviewed in [11]). It is widely accepted that a capacitation medium should contain a protein source, which is usually bovine serum albumin (BSA), and a variety of ions, including bicarbonate (HCO_3_^−^) and Ca^2+^ [5,12,13]. During capacitation, removal of cholesterol from sperm plasma membrane is induced by BSA, which leads to the displacement and rearrangement of phospholipids, increases membrane fluidity, primes acrosome exocytosis and prepares sperm for the penetration of oocyte vestments (including zona pellucida) prior to gamete fusion [14,15,16,17]. Furthermore, high levels of bicarbonate destabilize the sperm plasma membrane and trigger a signaling cascade that involves a sperm-specific soluble adenylyl cyclase (sAC), the increase in cAMP levels and the activation of protein kinase A (PKA; [18]). The activation of PKA is one of the most important signals that elicits sperm capacitation and is also involved in triggering acrosome exocytosis [1,19,20]. Additionally, the concentration of bicarbonate changes across epididymal regions and in the oviductal fluid [19]. These variations play a crucial role in the modulation of sperm function, as they are involved in events such as sperm maturation during the epididymal transit, the acquisition of motility upon ejaculation and the modulation of capacitation timing within the female reproductive tract [19]. After ejaculation, sperm encounter a high bicarbonate concentration in the female tract, ranging between 35 mM and 90 mM [5,21]. In vitro, these variations in the concentration of bicarbonate do not exist, but rather sperm are exposed to a standard concentration throughout all the process [22], this scenario likely being one of the limiting factors in the efficiency of in vitro capacitation.

Previous research used different concentrations of bicarbonate and BSA to elicit in vitro capacitation in pig sperm. Whereas several studies used a medium containing 25 mM NaHCO_3_ and 0.4% BSA [23], others used 36–38 mM NaHCO_3_ and 0.5% of BSA [9,24,25,26,27,28]. In addition, other authors even found that a medium with 5 mg/mL BSA but without NaHCO_3_ is able to induce in vitro capacitation in pig sperm [29,30]. These differences clearly indicate that while both BSA and NaHCO_3_ are important in the achievement of in vitro capacitation in pig sperm, neither the precise role of these two effectors nor their particular relevance in the achievement of sperm capacitation are well known. Since, to the best of our knowledge, no previous study interrogated the specific importance of both exogenous bicarbonate and BSA during the achievement of in vitro capacitation, this work aimed to explain to which extent these two components are crucial for pig sperm to elicit in vitro capacitation and undergo progesterone-induced acrosome exocytosis.

## 2. Materials and Methods

### 2.1. Reagents

Unless otherwise declared, all the reagents were purchased from Sigma–Aldrich (Saint-Louis, MO, USA) and Merck (Darmstadt, Germany). 

### 2.2. Semen Samples

Semen samples were obtained from 12 separate boars of proven fertility according to the farm records. Ejaculates were purchased from a local farm (Servicios Geneticos Porcinos, S.L., Roda de Ter, Spain), where boar semen was routinely collected twice a week through the gloved-hand method. Sperm-rich fractions were diluted in a commercial extender (Duragen^®^; Magapor, S.L.; Ejea de los Caballeros, Spain) to a final concentration of 3.3 × 10^7^ sperm/mL. Samples were cooled to 17 °C, packed into 90 mL doses and finally transported in an insulated container at 17 °C for approximately 45 min, which was the time required to arrive at the laboratory. Upon arrival, sperm quality was evaluated to confirm that conventional spermiogram parameters were above quality standards.

All procedures involving animals were performed according to the EU Directive 2010/63/EU for animal experiments, the Animal Welfare Law issued by the Regional Government of Catalonia, and the current regulation on Health and Biosafety issued by the Department of Agriculture, Livestock, Food and Fisheries, Regional Government of Catalonia (Spain). In addition, the production of seminal doses by the farm followed the ISO certification (ISO-9001:2008). The authors of this study did not manipulate any animal but rather purchased the semen samples from a local farm, which operates under commercial standard conditions. For this reason, no approval from a local ethics committee was required.

### 2.3. In Vitro Capacitation and Progesterone-Induced Acrosome Exocytosis

All media were freshly prepared on the day of use and consisted of Tyrode’s medium containing lactate and pyruvate, and different concentrations of bicarbonate and BSA. The composition of this basic medium was: 20 mM 4-(2-hydroxyethyl)-1-piperazineethanesulfonic acid (HEPES) buffer, 112 mM NaCl, 3.1 mM KCl, 5 mM glucose, 0.3 mM Na_2_HPO_4_, 0.4 mM MgSO_4_, 4.5 mM CaCl_2_, 21.7 mM sodium lactate and 1 mM sodium pyruvate. Eight different treatments were tested: (1) negative control (No BSA/No Bic), which was the basic medium without bicarbonate or BSA; (2) basic medium supplemented with 5 mg/mL BSA (BSA/No Bic); (3) basic medium supplemented with 5 mg/mL BSA and 5 mM NaHCO_3_ (BSA + 5 mM Bic); (4) basic medium supplemented with 5 mM NaHCO_3_ (No BSA + 5 mM Bic); (5) basic medium supplemented with 5 mg/mL BSA and 15 mM NaHCO_3_ (BSA + 15 mM Bic); (6) basic medium supplemented with 15 mM NaHCO_3_ (No BSA + 15 mM Bic); (7) basic medium supplemented with 5 mg/mL BSA and 38 mM NaHCO_3_ (BSA + 38 mM Bic); and (8) basic medium supplemented with 38 mM NaHCO_3_ (No BSA + 38 mM Bic). In all cases, pH was previously adjusted to 7.4 and media were equilibrated at 38.5 °C and 5% CO_2_ for 60 min before use. 

Seminal doses were separated from the diluent through centrifugation at 600× *g* and 17 °C for 10 min, washed with PBS, and resuspended with the previously described media to a final concentration of 2 × 10^7^ sperm/mL. Sperm were then incubated at 38.5 °C and 5% CO_2_ in humidified air and evaluated after 0 min, 120 min and 240 min of incubation, since 240 min has been reported to be enough time for pig sperm to become in vitro capacitated [16,30]. After 240 min of incubation, progesterone was added to a final concentration of 10 μg/mL to induce acrosome exocytosis [31,32]. In order to evaluate the response of capacitated sperm to progesterone, especially with regard to acrosome exocytosis, samples were further incubated under the same conditions and evaluated after 5 min, 30 min and 60 min of progesterone addition. At each relevant time point, separate aliquots were taken for the assessment of sperm motility, plasma membrane integrity, membrane lipid disorder, intracellular calcium levels, mitochondrial membrane potential and O_2_ consumption. Another aliquot was also taken for protein extraction and analysis. In this latter case, aliquots were centrifuged at 2400× *g* and 17 °C for 5 min and supernatants were discarded. Pellets were then stored at −80 °C until protein extraction.

### 2.4. Sperm Motility

Evaluation of sperm motility was performed using a CASA system (Integrated Sperm Analysis System V1.0; Proiser, Valencia, Spain). Briefly, 5 µL of each sample was placed onto a Makler counting chamber (Sefi-Medical Instruments). Samples were observed under a negative phase-contrast microscope (Olympus BX41; Olympus, Hamburg, Germany) at 100× magnification. Three replicates with a minimum of 1000 sperm per replicate were evaluated before calculating the corresponding mean ± SEM. The recorded sperm motility parameters were those described in Yeste et al. [33], namely: curvilinear velocity (VCL), which is the mean path velocity of the sperm head along its actual trajectory (μm/s); straight-line velocity (VSL), which is the mean path velocity of the sperm head along a straight line from its first to its last position (μm/s); average path velocity (VAP), which is the mean velocity of the sperm head along its average trajectory (μm/s); linearity coefficient (LIN), (VSL/VCL) × 100 (%); straightness coefficient (STR): (VSL/VAP) × 100 (%); wobble coefficient (WOB): (VAP/VCL) × 100 (%); mean amplitude of lateral head displacement (ALH), which is the mean value of the extreme side-to-side movement of the sperm head in each beat cycle (μm); and frequency of head displacement (BCF), which is the frequency with which the actual sperm trajectory crosses the average path trajectory (Hz). Total motility was defined as the percentage of sperm showing VAP > 10 μm/s, and progressive motility as the percentage of sperm exhibiting STR > 45%.

### 2.5. Flow Cytometry Analyses

Information about flow cytometry analyses is provided according to the recommendations of the International Society for Advancement of Cytometry (ISAC; [34]). Flow cytometry was used to evaluate plasma membrane integrity, membrane lipid disorder, acrosome integrity, mitochondrial membrane potential, and intracellular calcium levels in all samples and time-points. Prior to evaluation, sperm concentration was adjusted to 1 × 10^6^ sperm/mL in a final volume of 0.5 mL [35]. Sperm were then stained with the appropriate combinations of fluorochromes, following the protocols described below. In all cases, a minimum of 10,000 events per replicate were evaluated through a Cell Lab QuantaSC Flow Cytometer (Beckman Coulter, Fullerton, CA, USA). Particles were excited through an argon ion laser (488 nm) set at a power of 22 mW and cell diameter/volume was directly measured employing the Coulter principle for volume assessment. The EV channel was periodically calibrated using 10-μm Flow-Check fluorospheres (Beckman Coulter) by positioning this size of the bead at channel 200 on the volume scale. To capture fluorochrome signals, three different optical filters were used: FL1 (green fluorescence)—Dichroic/Splitter, DRLP: 550 nm, BP filter: 525 nm, detection width 505–545 nm; FL2 (orange fluorescence)—DRLP: 600 nm, BP filter: 575 nm, detection width: 560–590 nm; FL3 (red fluorescence)—LP filter: 670 nm, detection width: 655–685 nm. Debris (particle diameter < 7 µm) and aggregates (particle diameter > 12 µm) were excluded from the analysis by gating out the particles based on EV/side scatter (SSC) plots. In all tests except SYBR14/PI, data were corrected following the procedure described by Petrunkina et al. [36]. Each assessment per sample and parameter was repeated three times in independent tubes and the corresponding mean ± SEM was calculated.

#### 2.5.1. Evaluation of Sperm Membrane Integrity

Plasma membrane integrity was evaluated using the LIVE/DEAD Sperm Viability Kit (SYBR14/PI) following the protocol set by Garner and Johnson [37]. For this purpose, sperm samples were incubated with SYBR14 (final concentration: 100 nM) at 38 °C in darkness for 10 min and then with propidium iodide (PI; final concentration: 10 µM) at the same temperature for 5 min. Three sperm populations were identified in flow-cytometry dot plots: (i) membrane-intact sperm (green-stained), which were positive for SYBR14 and negative for PI (SYBR14^+^/PI^−^); (ii) non-viable sperm (red-stained), which were negative for SYBR14 and positive for PI (SYBR14^−^/PI^+^); and (iii) non-viable sperm (orange-stained), which were stained both in green and red (SYBR14^+^/PI^+^). 

#### 2.5.2. Evaluation of Sperm Membrane Lipid Disorder

Lipid disorder of plasma membrane was determined with Merocyanine 540 (M540) and YO-PRO-1, following the procedure set by Harrison et al. [38]. Briefly, sperm were stained with M540 (final concentration: 2.6 µM) and YO-PRO-1 (final concentration: 25 nM) and incubated at 38 °C for 10 min in the dark. A total of four sperm populations were identified: (i) non-viable sperm with low membrane lipid disorder (M540^−^/YO-PRO-1^+^); (ii) non-viable sperm with high membrane lipid disorder (M540^+^/YO-PRO-1^+^); (iii) viable sperm with low membrane lipid disorder (M540^−^/YO-PRO-1^−^); and (iv) viable sperm with high membrane lipid disorder (M540^+^/YO-PRO-1^−^). 

#### 2.5.3. Evaluation of Acrosome Integrity

Acrosome integrity was determined through staining with the lectin from *Arachis hypogaea* (peanut agglutinin, PNA) conjugated with fluorescein isothiocyanate (FITC) and ethidium homodimer (3,8-diamino-5-ethyl-6-phenylphenanthridinium bromide; EthD-1), following the protocol described by Rocco et al. [9]. Samples were incubated with EthD-1 (final concentration: 2.5 µg/mL) at 38 °C for 5 min in the dark. Following centrifugation at 2000× *g* and 17 °C for 30 s, sperm were resuspended with PBS containing 4 mg/mL BSA. Samples were again centrifuged under the same conditions and subsequently added to 100 µL of ice-cold methanol (100%) for 30 s. After centrifugation at 2000× *g* and 17 °C for 30 s, samples were resuspended with 250 µL PBS before adding 0.8 µL PNA-FITC (final concentration: 2.5 µM). Samples were incubated at 25 °C for 15 min in the dark and then centrifuged at 2000× *g* and 17 °C for 30 s. Pellets were resuspended with 0.6 µL PBS and centrifuged at 2000× *g* and 17 °C for 30 s. This step was repeated twice. Sperm were evaluated with the flow cytometer and four sperm populations were identified: (i) viable sperm with an intact acrosome (PNA^+^/EthD-1^−^); (ii) viable sperm with an exocytosed acrosome (PNA^−^/EthD-1^−^); (iii) non-viable sperm with an intact acrosome (PNA^+^/EthD-1^+^); and (iv) non-viable sperm with an exocytosed acrosome (PNA^−^/EthD-1^+^). Fluorescence of EthD-1 was detected through FL3, whereas that of PNA was detected through FL1. 

#### 2.5.4. Evaluation of Intracellular Calcium Levels

Intracellular calcium levels were evaluated through two different fluorochromes (Fluo3 and Rhod5; [7]). For Fluo3, sperm were incubated at 38 °C in the dark for 10 min with Fluo3-AM (final concentration: 1 µM) and PI (final concentration: 12 µM), which were detected through FL1 and FL3, respectively. Four sperm populations were identified: (i) viable sperm with low levels of intracellular calcium (Fluo3^−^/PI^−^), (ii) viable sperm with high levels of intracellular calcium (Fluo3^+^/PI^−^), (iii) non-viable sperm with low levels of intracellular calcium (Fluo3^−^/PI^+^), and (iv) non-viable sperm with high levels of intracellular calcium (Fluo3^+^/PI^+^). The geometric mean of Fluo3 intensity was recorded for all sperm populations.

With regard to Rhod5, sperm were incubated at 38 °C for 10 min in the dark with Rhod5-N (final concentration: 5 µM) and YO-PRO-1 (final concentration: 25 nM). Filters used to detect the fluorescence from Rhod5 and YO-PRO-1 were FL3 and FL1, respectively. A total of four sperm populations were identified: (i) viable sperm with low levels of intracellular calcium (Rhod5^−^/YO-PRO-1^−^), (ii) viable sperm with high levels of intracellular calcium (Rhod5^+^/YO-PRO-1^−^), (iii) non-viable sperm with low levels of intracellular calcium (Rhod5^−^/YO-PRO-1^+^), and (iv) non-viable sperm with high levels of intracellular calcium (Rhod5^+^/YO-PRO-1^+^). The geometric mean of Rhod5 intensity was recorded for all sperm populations.

#### 2.5.5. Evaluation of Mitochondrial Membrane Potential

Mitochondrial membrane potential (MMP) was determined through JC1 (5,5′,6,6′-tetrachloro-1,1′,3,3′-tetraethylbenzimidazolylcarbocyanine iodide) fluorochrome, following the protocol described by Guthrie and Welch [39]. For this purpose, sperm samples were incubated with JC1 (final concentration: 0.3 µM) at 38 °C in the dark for 30 min, and two sperm populations were distinguished: (i) sperm with high mitochondrial membrane potential, and (ii) sperm with low mitochondrial membrane potential. When MMP was high, JC1 inside mitochondria formed orange aggregates that were detected through FL2. Ratios between orange (FL2) and green (FL1) fluorescence in the sperm population with high MMP (JC1_agg_) were also calculated. 

### 2.6. Sodium Dodecyl Sulfate Polyacrylamide Gel Electrophoresis (SDS–PAGE)

Sperm pellets stored at −80 °C were thawed and resuspended in 400 µL ice-cold lysis buffer and maintained at 4 °C for 30 min under constant agitation. The lysis buffer was made up of 2% SDS, 1% Triton-X-100, 8 M urea, 2 mM dithiothreitol (DDT), 0.5% Tween 20 and 50 mM Tris-HCl; the pH was adjusted to 7.4. On the day of use, the lysis buffer was mixed with 1% commercial protease inhibitor cocktail (Sigma–Aldrich), 1% phenylmethanesulfonyl fluoride (PMSF), and 0.15% sodium orthovanadate. Following this, samples were homogenized by sonication (50% amplitude; 10 long-lasting pulses; Bandelin Sonopuls HD 2070; Bandelin Electronic GmbH and Co., Heinrichstrasse, Berlin), and then centrifuged at 10,000× *g* and 4 °C for 15 min. Supernatants were carefully collected and total protein was quantified in triplicate with a detergent-compatible protein assay (DC Protein Assay; BioRad, Hercules, CA, USA). Standard curves were produced with different concentrations of BSA (Quick Start Bovine Serum Albumin Standard; Bio-Rad).

For SDS–PAGE separation, 10 µg protein of each sample was diluted with 2× Laemmli reducing buffer containing 5% (*v*:*v*) β-mercaptoethanol (Bio-Rad Laboratories). Samples were boiled at 90 °C for 5 min and then loaded onto 1-mm SDS–PAGE gels, together with a molecular-weight marker (All Blue Precision Plus Protein Standards; Bio-Rad Laboratories). The separating gel contained 12% (*w*:*v*) acrylamide, whereas the stacking gel contained 5% (*w*:*v*) (Bio-Rad Laboratories). Subsequently, gel electrophoresis was run at constant voltage (80 V for stacking gels and 120 V for separating gels). Proteins were transferred onto polyvinylidene fluoride membranes (Immobilon-P; Millipore, Darmstadt, Germany) at 120 mA for 120 min. Thereafter, membranes were incubated with blocking solution, consisting of 5% BSA (*w*:*v*) in TBST at 4 °C overnight with agitation. 

### 2.7. Immunoblotting of Tyr-P-GSK3

Membranes were incubated with a primary anti-phospo-GSK3 (Tyr279/Tyr216) antibody (Tyr-P-GSK3; clone 5G-2F, 05- 413, Millipore) diluted (1:1000; *v*:*v*) in blocking solution at room temperature and agitation for 60 min. After washing membranes four times (5 min each) in TBST, they were incubated with a secondary horseradish peroxidase (HRP)-conjugated anti-mouse antibody (P0260; Dako Denmark A/S, Glostrup, Denmark) diluted 1:5000 (*v*:*v*) in blocking solution at room temperature with agitation for 60 min. Membranes were subsequently washed eight times (5 min each) in TBST. 

Reactive bands were visualized with a chemiluminescent substrate (Immobilon Western Chemiluniniscent HRP Substrate; Millipore) and scanned using G:BOX Chemi XL 1.4 (SynGene, Frederick, MT, USA) and GeneSys image acquisition software v1.2.8.0 (SynGene). Protein bands from scanned images were quantified through Quantity One v4.6.9 software package (Bio-Rad Laboratories, Inc.). Protein quantification was expressed as the total signal intensity inside the boundary of a band measured in pixel intensity units (density, square millimeter) minus the background signal, considered as 0 (white). 

Data were normalized with α-tubulin following the stripping of the same membranes. In brief, membranes were stripped through two incubations at room temperature under agitation for 10 min with a stripping buffer containing 0.02 M glycine, 3% (*w*:*v*) SDS and 1% (*v*:*v*) Tween 20 (pH 2.2) in 100 mL Mili-Q water. Thereafter, membranes were washed twice in TBS at room temperature for 10 min and twice with TBST for 5 min. Following this, stripped membranes were incubated with an anti-α-tubulin mouse antibody (anti-α-tubulin antibody clone DM1A, MABT205; Millipore) diluted 1:5000 (*v*:*v*) in blocking solution at room temperature under agitation for 60 min. After washing with blocking solution twice (5 min each), membranes were incubated with a HRP-conjugated anti-mouse antibody (P0260; Dako Denmark A/S) diluted 1:15,000 (*v*:*v*) in blocking solution (*v*:*v*) at room temperature under agitation for 60 min. Reactive bands were visualized and band density calculated as previously described. Relative content of Tyr-P-GSK3α was normalized, dividing the intensity of Tyr-P-GSK3α by that of α-tubulin. Data were corrected to a basal arbitrary value of 100 for the control point, which corresponded to the incubation in the medium containing BSA and 38 mM bicarbonate at 0 min.

### 2.8. Immunoblotting of DARPP-32 (PKA)

The procedure was similar to that carried out to determine the Tyr-P-GSK3α/α-tubulin ratio. Thus, after the SDS-PAGE/membrane transference procedure conducted as described above, membranes were incubated with an anti-phospho-DARPP-32 (Thr75) primary antibody (Cell Signaling Technology; Leyden; The Netherlands) at 4 °C overnight. The dilution factor was 1:1000 (*v*:*v*). After three washes, membranes were incubated with a horseradish peroxidase (HRP)-conjugated donkey anti-rabbit secondary antibody (Dako; Glostrup, Denmark) at a dilution of 1:5000 (*v*:*v*) in blocking solution for 60 min. Membranes were washed six times and then revealed using a chemiluminescent HRP substrate (ImmunoCruz Western Blotting Luminol Reagent; Santa Cruz Biotechnology, Dallas, TX, USA). Protein quantification was expressed as the total signal intensity inside the boundary of a band measured in pixel intensity units (density, square millimeter) minus the background signal, considered as zero (white). Afterwards, the same membranes were reprobed with a specific total PKAα antibody. For this purpose, membranes were stripped through two incubations at room temperature under agitation for 10 min with a stripping buffer containing 0.02 M glycine, 3% (*w*:*v*) SDS and 1% (*v*:*v*) Tween 20 (pH 2.2) in 100 mL Mili-Q water. Afterwards, membranes were incubated with the specific anti-PKAα primary antibody (Cell Signaling Technology; final dilution: 1:1000, *v*:*v*) at 4 °C overnight. After three washes, membranes were incubated with a horseradish peroxidase (HRP)-conjugated donkey anti-rabbit secondary antibody (Dako; Glostrup, Denmark) at a dilution of 1:5000 (*v*:*v*) in blocking solution for 60 min. Membranes were washed six times and were then revealed using a chemiluminescent HRP substrate (ImmunoCruz Western Blotting Luminol Reagent; Santa Cruz Biotechnology, Dallas, TX, USA). Reactive bands were visualized and their densities were calculated as previously described. At this point, the DARPP-32/PKA ratio was calculated by dividing the intensity of DARPP-32 by that of PKA. Data were corrected to a basal arbitrary value of 100 for the control point, which corresponded to the incubation in the medium containing BSA and 38 mM bicarbonate at 0 min.

### 2.9. Determination of O_2_ Consumption Rate

Determination of O_2_ consumption rate was carried out through a SensorDish^®^ Reader (SDR) system (PreSens Gmbh; Regensburg, Germany). Briefly, 1 mL of each sample was incubated in four separate media (No BSA/No Bic, BSA/No Bic, BSA + 38 mM Bic, and No BSA + 38 mM Bic) for 0 min or 240 min. Following this, samples were transferred onto Oxodish^®^ OD24 plates (24 wells/plate) specifically designed for this device. Plates were sealed with Parafilm^®^, loaded into the SDR system, and incubated at 38.5 °C (controlled atmosphere) for 120 min. During that period, O_2_ concentration was recorded in each well at a rate of one reading/min. Results were exported to an Excel file and O_2_ consumption rate was normalized against the total number of viable sperm per sample, determined through flow cytometry (SYBR14^+^/PI^−^ sperm) using another aliquot. For this experiment, only a single concentration of bicarbonate (38 mM) was used. This was because, from an operative point of view, the analysis of all bicarbonate concentrations at all incubation times was not feasible. Thus, the authors decided to focus on the highest concentration (38 mM) and on the outcomes after 240 min of incubation at 38.5 °C and at a 5% CO_2_ atmosphere, as this is the time needed for pig sperm to achieve the capacitated status.

### 2.10. Statistical Analyses

Data were analyzed with a statistical package (IBM SPSS for Windows, ver. 25.0; Armonk, NY, USA) and are shown as mean ± standard error of the mean (SEM). Data were first checked for normality (Shapiro–Wilk test) and homogeneity of variances (Levene test). Following this, a linear mixed model (repeated measures) was run with the treatment as the fixed-effects inter-subject factor and the incubation time as the intra-subjects factor. Sidak’s post-hoc test was run for pair-wise comparisons. The level of significance was set at *p* ≤ 0.05.

## 3. Results

### 3.1. Effects of Different Concentrations of Bicarbonate and BSA on Plasma Membrane Integrity

Plasma membrane integrity during in vitro capacitation and after the addition of progesterone was evaluated through SYBR14/PI, as shown in Figure 1 (mean ± SEM). A significant (*p* < 0.05) reduction in the percentages of sperm with an intact plasma membrane (SYBR14^+^/PI^−^) was observed in treatments containing 38 mM of bicarbonate, with or without BSA, after 120 min of starting the experiment. After 120 min and 240 min of incubation, all media containing either no bicarbonate or bicarbonate at 5 mM presented significantly (*p* < 0.05) higher percentages of sperm with an intact plasma membrane than treatments containing 15 mM or 38 mM bicarbonate. After 300 min of incubation, the medium without BSA and without bicarbonate had a significantly (*p* < 0.05) higher percentage of sperm with an intact plasma membrane than the other treatments.

### 3.2. Effects of Different Concentrations of Bicarbonate and BSA on Membrane Lipid Disorder 

The impact of different bicarbonate and BSA concentrations on the lipid disorder of sperm membrane is shown in Figure 2. Percentages of viable sperm with low lipid membrane disorder (M540^−^/YO-PRO-1^−^; Figure 2a) after 120 min of incubation were significantly higher (*p* < 0.05) in the treatments without bicarbonate, either with or without BSA, than in those containing bicarbonate at 5 mM, 15 mM and 38 mM. Additionally, percentages of viable sperm with low membrane lipid disorder in the medium without BSA and with bicarbonate at 38 mM were significantly (*p* < 0.05) lower than in the other treatments throughout the entire period of incubation.

On the other hand, when percentages of sperm with high membrane lipid disorder (M540^+^) were calculated considering the viable sperm population only (YO-PRO-1^−^), the media containing BSA, with or without bicarbonate, showed significantly (*p* < 0.05) higher values of this parameter than those without BSA, from 240 min and until the end of the incubation period (Figure 2b). Remarkably, treatments containing BSA and bicarbonate at 5 mM, 15 mM or 38 mM showed significantly (*p* < 0.05) higher percentages of M540^+^/viable sperm than those containing BSA and no bicarbonate throughout all the incubation time.

### 3.3. Effects of Different Concentrations of Bicarbonate and BSA on Acrosome Integrity

As shown in Figure 3a, percentages of viable sperm with an intact acrosome (PNA^+^/EthD-1^−^) in the medium containing BSA and 38 mM were significantly (*p* < 0.05) lower than in the other treatments throughout the entire incubation period. In contrast, media without bicarbonate, either with or without BSA, showed significantly (*p* < 0.05) higher percentages of viable sperm with an intact acrosome than the other treatments. In addition, it was observed that the higher the concentration of bicarbonate, the lower the percentages of viable sperm with an intact acrosome. With regard to the percentages of viable sperm with an exocytosed acrosome (PNA^−^/EthD-1^−^), the presence of BSA in the media containing the same concentration of bicarbonate (i.e., 0 mM, 5 mM, 15 mM and 38 mM) led to significantly (*p* < 0.05) higher percentages of this sperm population after 240 min, 245 min, 270 min and 300 min of incubation (Appendix A).

When percentages of sperm with an exocytosed acrosome (PNA^−^) were calculated considering the viable sperm population (EthD-1^−^), no significant differences between treatments were observed at 0 min and 120 min (Figure 3b). However, treatments containing BSA, with or without bicarbonate, showed significantly (*p* < 0.05) higher percentages of viable sperm with an exocytosed acrosome (PNA^−^/viable sperm) than treatments without BSA, right after the addition of progesterone (245 min), and after 270 min and 300 min of incubation. In addition, percentages of viable sperm with an exocytosed acrosome were significantly (*p* < 0.05) higher in the medium with BSA and 38 mM than in that with BSA but without bicarbonate, from 245 min to 300 min of incubation. 

### 3.4. Effects of Different Concentrations of Bicarbonate and BSA on Intracellular Calcium Levels

Percentages of viable sperm with high intracellular calcium levels (%Fluo3^+^/PI^−^ and %Rhod5^+^/YO-PRO-1^−^ sperm) are shown in Figure 4. Significantly (*p* < 0.05) higher percentages of Fluo3^+^/PI^−^ sperm were observed in the medium containing BSA and 38 mM bicarbonate from 0 min to 300 min of incubation, with maximum values being observed after 5 min of progesterone addition (i.e., 245 min; Figure 4a). On the other hand, the medium without BSA and with 5 mM bicarbonate showed the lowest percentage of Fluo3^+^/PI^−^ sperm when compared to the other experimental conditions.

In a similar fashion to that observed for Fluo-3/PI staining, percentages of viable sperm with a positive Rhod5 signal (Rhod5^+^/YO-PRO-1^−^) progressively increased when sperm were incubated in media containing BSA (*p* < 0.05), especially at 38 mM bicarbonate (Figure 4b). The addition of progesterone induced a rapid increase in these percentages, which reached the highest values 5 min after progesterone addition (i.e., 245 min of incubation). Moreover, the lowest percentages of Rhod5^+^/YO-PRO-1^−^ sperm from 0 min to 300 min of incubation were observed in the media without BSA, especially when containing 5 mM, 15 mM and 38 mM bicarbonate.

### 3.5. Effects of Different Concentrations of Bicarbonate and BSA on Mitochondrial Membrane Potential

Incubation of pig spermatozoa with BSA and 38 mM bicarbonate significantly (*p* < 0.05) increased the percentage of spermatozoa with high mitochondrial membrane potential (MMP) from 0 min to 300 min of incubation (Figure 5a). The addition of progesterone after 240 min of incubation induced a rapid increase in this percentage, which was followed by a decrease at 270 min. On the other hand, while incubation of sperm in a medium without BSA and without bicarbonate (No BSA/No Bic) prevented the increase in the percentage of sperm with high MMP from 0 min to 240 min of incubation, the addition of progesterone to this medium after 240 min of incubation also increased that percentage (Figure 5a). Remarkably, percentages of sperm with high MMP did not differ between the medium without BSA/bicarbonate (No BSA/No Bic) and the medium with BSA and 38 mM bicarbonate (BSA + 38 mM Bic) after 270 min of incubation. At 300 min, sperm incubated with BSA and 38 mM bicarbonate showed a significantly (*p* < 0.05) higher percentage of sperm with high MMP than in the other experimental conditions. Furthermore, the addition of progesterone to the medium with BSA and 15 mM bicarbonate after 240 min of incubation also showed an increased percentage of sperm with high MMP. However, the extent of that increase was significantly (*p* < 0.05) lower than that observed in the medium containing BSA and 38 mM bicarbonate. Media without BSA showed, in all cases, significantly (*p* < 0.05) lower percentages of sperm with high MMP than those containing both BSA and bicarbonate (Figure 5a). 

Figure 5b shows the FL2:FL1 ratio in the sperm population with high MMP. After 240 min of incubation, these ratios were significantly (*p* < 0.05) higher in the media with BSA than in those without this protein, especially when exogenous bicarbonate was present. Following the addition of progesterone, the treatments containing BSA and bicarbonate at 15 mM (BSA + 15 mM Bic) or 38 mM (BSA + 38 mM Bic) showed significantly (*p* < 0.05) higher FL2:FL1 ratios than the other media. In spite of this, after 270 min of incubation, the media containing BSA either with (38 mM) or without bicarbonate, showed significantly (*p* < 0.05) higher FL2:FL1 ratios than the media without BSA. At 300 min, no significant differences between treatments were observed (Figure 5b).

### 3.6. Effects of Different Concentrations of Bicarbonate and BSA on Sperm Motility

In all media, total sperm motility decreased throughout incubation time, reaching minimal values at the end of the experiment (Figure 6a). In addition, percentages of total motile spermatozoa incubated in media containing BSA and bicarbonate at either 0 mM or 5 mM were significantly (*p* < 0.05) higher than in the other treatments. In contrast, incubation of spermatozoa in the medium with 38 mM bicarbonate and no BSA (No BSA + 38 mM Bic) showed the lowest percentage of total motile sperm, with almost complete immobilization at the end of the experiment (Figure 6a). Furthermore, sperm agglutination was very high in treatments containing 38 mM, either with or without BSA, from 120 min and 300 min of incubation.

Similar results were observed for the percentages for progressively motile spermatozoa (Figure 6b). Again, the medium with BSA and 5 mM bicarbonate presented the highest progressive sperm motility, followed by the medium with BSA and without bicarbonate. In contrast, the medium containing 38 mM bicarbonate, either with or without BSA, showed the lowest percentages of progressively motile sperm.

As shown in Table 1, VCL was significantly (*p* < 0.05) higher in treatments containing BSA than in those that did not contain this protein after 240 min, 245 min, 270 min and 300 min of incubation. Although the treatment with BSA and the highest concentration of bicarbonate (i.e., 38 mM) showed the highest VCL after 240 min, 245 min, 270 min and 300 min of incubation, significant (*p* < 0.05) differences were only observed when the medium with BSA and without bicarbonate was compared with that containing BSA and 38 mM bicarbonate. On the other hand, sperm incubated in media with BSA, regardless of whether or not they contained bicarbonate, showed significantly (*p* < 0.05) higher VSL and VAP than those incubated without BSA, from 240 min to 300 min of incubation (Table 1).

Percentages of LIN were significantly (*p* < 0.05) lower in the medium without BSA and without bicarbonate than in those with BSA and/or different concentrations of bicarbonate after 240 min of incubation. At 270 min and 300 min, percentages of LIN were significantly (*p* < 0.05) lower in the treatment containing BSA and 38 mM bicarbonate than in that containing BSA and no bicarbonate (Table 2). With regard to percentages of STR, media that did not contain BSA and 0 mM (No BSA/No Bic) or 15 mM bicarbonate (No BSA + 15 mM Bic) showed significantly (*p* < 0.05) lower values of this parameter than their counterparts containing BSA (BSA/No Bic and BSA + 15 mM Bic) after 120 min of incubation. From 240 min to 300 min of incubation, percentages of STR were significantly (*p* < 0.05) lower in the treatment without BSA and without bicarbonate (No BSA/No Bic) than in that containing BSA and 5 mM bicarbonate (BSA + 5 mM Bic). Finally, percentages of WOB were significantly (*p* < 0.05) higher in the medium without BSA and with 5 mM bicarbonate (No BSA + 5 mM Bic) than in the other treatments after 120 min of incubation. After 245 min and 270 min of incubation, percentages of WOB were significantly (*p* < 0.05) higher in sperm incubated with BSA and 5 mM bicarbonate than in those incubated without BSA and the same concentration of bicarbonate (i.e., BSA + 5 mM Bic vs. No BSA + 5 mM Bic). At 300 min, while treatments containing BSA and bicarbonate at 5 mM or 15 mM showed significantly (*p* < 0.05) higher percentages of WOB than those with no BSA and the same concentrations of bicarbonate (i.e., BSA + 5 mM Bic vs. No BSA + 5 mM Bic; BSA + 15 mM Bic vs. No BSA + 15 mM Bic), those observed in the medium containing BSA and 38 mM bicarbonate were significantly (*p* < 0.05) lower than in those with no BSA and bicarbonate at 38 mM (i.e., BSA + 38 mM Bic vs. No BSA + 38 mM Bic). 

Table 3 shows ALH and BCF of pig sperm incubated with different media. With regard to ALH, although no significant differences between treatments were observed at the beginning of the experiment (0 min), those containing BSA and bicarbonate (5 mM, 15 mM or 38 mM) showed significantly (*p* < 0.05) higher values of this parameter than those without BSA, and the same concentration of bicarbonate (BSA + 5 mM Bic vs. No BSA + 5 mM Bic; BSA + 15 mM Bic vs. No BSA + 15 mM Bic; BSA + 38 mM Bic vs. No BSA + 38 mM Bic) after 120 min of incubation and until the end of the experimental period. Finally, sperm incubated with BSA, regardless of the concentration of bicarbonate, showed significantly (*p* < 0.05) higher BCF than those incubated without BSA after 245 min, 270 min and 300 min of incubation. 

### 3.7. Effects of Different Concentrations of Bicarbonate and BSA on DARPP-32 Phosphorylation Levels 

Figure 7 shows normalized phosphorylated DARPP-32:PKA ratios after 0 min, 240 min, 245 min and 300 min of incubation in media, with or without BSA, and with or without 38 mM bicarbonate. Only these four treatments and time points are shown, since they were those in which differences were clearer. Sperm incubated in media containing either BSA alone (BSA/No Bic) or BSA and 38 mM bicarbonate (BSA + 38 mM Bic) showed significantly (*p* < 0.05) higher phosphorylated DARPP-32:PKA ratios than the other treatments after 0 min, 240 min and 245 min of incubation (Figure 7a). Remarkably, sperm incubated in the medium containing 38 mM bicarbonate but no BSA (No BSA + 38 mM Bic) showed similar phosphorylated DARPP-32:PKA ratios to those incubated in a medium without BSA and without bicarbonate (No BSA/No Bic) after 0 min and 240 min of incubation. At 300 min, the medium without BSA and with 38 mM bicarbonate (No BSA + 38 mM Bic) showed a significantly (*p* < 0.05) lower phosphorylated DARPP-32:PKA ratio than the other treatments.

### 3.8. Effects of Different Concentrations of Bicarbonate and BSA on Tyrosine Phosphorylation Levels of GSK3α 

Figure 8 shows phosphorylated GSK3α:α-tubulin ratios after 0 min, 240 min, 245 min and 300 min of incubation in media, with or without BSA, and with or without 38 mM bicarbonate. In a similar fashion to the case of phosphorylated DARPP-32:PKA ratios, only these four treatments and time points are shown, as they were those in which differences were clearer. Incubation of sperm in media containing BSA and/or 38 mM bicarbonate (i.e., BSA/No Bic, BSA + 38 mM, and No BSA + 38 mM) showed significantly (*p* < 0.05) higher phosphorylated GSK3α:α-tubulin ratios after 0 min and 240 min of incubation. At 245 min, sperm incubated in a medium with BSA and 38 mM bicarbonate presented significantly (*p* < 0.05) higher phosphorylated GSK3α:α-tubulin ratios than the other three treatments, and those incubated in a medium with BSA and without bicarbonate exhibited significantly (*p* < 0.05) higher phosphorylated GSK3α:α-tubulin ratios than those incubated in media without BSA (No BSA/No Bic, No BSA + 38 mM Bic). At 300 min, phosphorylated GSK3α:α-tubulin ratios in sperm incubated in media containing BSA, with or without bicarbonate, were significantly (*p* < 0.05) higher than in those incubated in media without BSA.

### 3.9. Effects of Bicarbonate and BSA on O_2_ Consumption Rate 

Figure 9 shows the O_2_ consumption rate in pig sperm incubated in media containing BSA and/or 38 mM bicarbonate (i.e., No BSA/No Bic, BSA/No Bic, BSA + 38 mM Bic, and No BSA + 38 mM Bic). 

At 0 min, the O_2_ consumption rate in sperm diluted in the medium without BSA and bicarbonate (No BSA/No Bic) was low (0.90 ± 0.18 µmoL/h × 10^6^ viable sperm). These ratios did not significantly increase after 240 min of incubation. Similar results were observed in the treatment containing 38 mM bicarbonate and no BSA (No BSA + 38 mM Bic). In contrast, sperm incubated in the presence of BSA, either with or without bicarbonate, showed significantly (*p* < 0.05) higher O_2_ consumption rates than those incubated without BSA, at both 0 min (2.06 ± 0.32 µmoL/h × 10^6^ viable sperm) and 240 min (2.77 ± 0.40 µmoL/h × 10^6^ viable sperm).

## 4. Discussion

Our findings clearly indicate that the presence of BSA is essential for pig sperm to elicit in vitro capacitation. Moreover, bicarbonate is not essential to launch the in vitro capacitation process, but it modulates its efficiency. Therefore, the current work supports that in vitro capacitation of pig sperm can be achieved in a medium without bicarbonate, but not without BSA, although the efficiency of the process is not optimal in the absence of the former. This conclusion is not only reached from the changes in sperm function parameters related to capacitation, such as membrane lipid disorder, acrosome exocytosis and intracellular calcium levels, but also from the increase observed in phosphorylation levels of Tyr-GSK3α and DARPP-32. In fact, previous works already demonstrated that in vitro capacitation in pig sperm can be efficiently achieved in a medium with BSA but without bicarbonate [7,29,30,32,40]. In this context, it is worth noting that these observations are somewhat different from those observed in other mammalian species, in which bicarbonate is considered to be an essential component to elicit sperm capacitation [41]. These discrepancies do highlight the existence of species-specific mechanisms that are regulating the achievement of sperm capacitation in vitro. In relation, while different studies were conducted to induce in vitro capacitation in pig sperm, no consensus was reached on what the best medium is. For instance, high concentrations of bicarbonate (i.e., 37.6 mM) in in vitro capacitation media induce strong agglutination in pig sperm [9], thereby greatly impairing their recovery. While high concentrations of bicarbonate also induce sperm agglutination in sheep [42] and horses [43], their impact is not so apparent in hamsters [41] and cattle [44]. In addition, experimental conditions also differ between studies evaluating the response of pig sperm to in vitro capacitation. In effect, whereas some studies, including the present one, incubated sperm cells for 240 min prior to adding progesterone [45,46,47], other authors used shorter incubation periods (60 min [22], 120 min [28], and 180 min [48,49]). The fact that the optimal concentration of bicarbonate in the in vitro capacitation medium of mammalian sperm differs between species should not be considered as exceptional, since, for example, other components such as heparin are required for bovine, but not for porcine sperm [44]. Therefore, although the general mechanism of mammalian sperm capacitation is known [5], it seems that species differ in particular events triggered by definite components. 

Focusing on the addition of bicarbonate to capacitation medium, the results of our work have elucidated some interesting aspects. With regard to membrane integrity and lipid disorder, capacitating media containing high concentrations of bicarbonate (15 mM and 38 mM) led to the lowest percentages of membrane-intact sperm and the highest percentages of viable sperm with high membrane lipid disorder. This was expected, since bicarbonate is known to be an important regulator of phospholipid scrambling, thereby affecting membrane fluidity [50]. In effect, high intracellular levels of bicarbonate induce plasma membrane asymmetry, which activates scramblase enzymes that translocate membrane phospholipids, such as phosphatidylserine and phosphatidylethanolamine, thereby reducing membrane stability and making cholesterol available to external receptors [50,51]. Cholesterol can then be removed by BSA, which leads to the rapid collapse of membrane asymmetry [5,12,52]. Based on our data, the addition of bicarbonate to capacitation medium does accelerate the increase in the percentage of sperm with high membrane lipid disorder, which takes more time when sperm are incubated without bicarbonate in an atmosphere of 5% CO_2_. The fact that pig sperm do not require exogenous bicarbonate to elicit in vitro capacitation can be related to their specific membrane structure. It is well known that the proportion of unsaturated:saturated fatty acids in pig sperm membrane is higher, and that of cholesterol:phospholipid lower, than in other species, such as the human, horse and cattle [53,54,55,56]. In this way, it is likely that the specific composition of pig sperm plasma membrane makes these cells more prone to increase their membrane lipid disorder in the presence of BSA under a 5% CO_2_ atmosphere, which would be sufficient to induce these capacitation-related changes. A concentration of bicarbonate as low as 2 mM has also been reported by other authors to induce in vitro capacitation in pig sperm [45].

The aforementioned changes in sperm plasma membrane appeared to be linked with the results observed in sperm motility. Sperm incubated in in vitro capacitation media containing BSA and either 5 mM or 15 mM bicarbonate showed higher total and progressive motility, and VCL, VSL and VAP. In addition, our sperm motility data agreed with the significant changes observed in the sperm population with high MMP, matching with previous works reporting a significant correlation between MMP and sperm motility [57,58,59]. It is well known that mitochondrial activity increases during in vitro capacitation [60]; this activity further raises after the induction of acrosome exocytosis, which is likely to be related to the increase in the rate of mitochondrial oxidative phosphorylation [7]. In addition, we observed a mitochondria-stored calcium peak, indicated by Fluo3-staining, immediately after the induction of acrosome exocytosis with progesterone. In this context, it is worth bearing in mind the close relationship between sperm calcium metabolism and mitochondrial function. Calcium is a key regulator for sperm hyperactivation in the mouse [61], which is an essential step in the achievement of sperm capacitation. This motile pattern is characterized by an asymmetrical and wide amplitude flagellar movement [62]. In our conditions, the highest concentration of bicarbonate led to the lowest values of both total and progressive motility, despite the existence of an acrosome exocytosis-related calcium peak. These results, combined with the observed changes in the percentages of both plasma membrane and acrosome integrity, suggest that high concentrations of bicarbonate cause damage to the sperm cell. Although these detrimental effects seem to be related to capacitation-linked changes, our results suggest that adding bicarbonate in the range of 30–40 mM should be avoided if pig sperm are capacitated under a 5% CO_2_ atmosphere for 240 min. 

In this study, we used two different fluorochromes (Fluo3 and Rhod5) that stain calcium stores of both sperm head and mid-piece [7]. It is well understood that in vitro capacitation and the subsequent progesterone-induced acrosome exocytosis increase calcium influx through CatSper channels, which is required for sperm to acquire the ability to fertilize the oocyte [63,64]. In pigs, previous research showed that calcium plays a vital role during sperm capacitation and acrosome reaction [7,65]. Herein, we observed that the presence of BSA in the capacitation medium is essential for the calcium peak to occur after the induction of acrosome exocytosis with progesterone. These data are in agreement with Espinosa et al. [66], who found that BSA in the medium is responsible for calcium to enter mouse spermatogenic cells. In addition, the calcium peak was proportional to bicarbonate concentration, with the highest peaks being observed at 38 mM, followed by 15 mM and 5 mM bicarbonate. These results indicate that the entry of calcium into the sperm is a result of the joint action of BSA and bicarbonate. As stated previously, removal of cholesterol from sperm plasma membrane during capacitation is induced by BSA, which leads to the displacement and rearrangement of phospholipids and increases membrane fluidity [67,68]. Cholesterol efflux triggers a cascade of events, such as the increase of calcium levels via CatSper channels [69] or the activation, through bicarbonate, of a soluble, non-membrane bound adenylyl cyclase (sAC) [5,70]. This latter point is important, since activation of sAC leads to an increase in intracellular cAMP levels, which in turn activate protein kinase A, a crucial protein for the signaling pathway of mammalian sperm capacitation [17,71]. The fact that low concentrations (5 mM) or even the complete absence of bicarbonate in in vitro capacitation media also allows pig sperm to increase their intracellular calcium levels suggests that, under a 5% CO_2_ atmosphere, the amount of bicarbonate required to activate sAC does not need to be high. 

On the other hand, we also analyzed the activity of PKA through tyrosine phosphorylation levels of DARPP-32. DARPP-32 is involved in the modulation of the activity of several capacitation-involved protein kinases and phosphatases [72]. The activity of DARPP-32 is regulated by a complex system of intracellular signaling pathways involving PKA and calcium [73,74]. Remarkably, our results showed that the highest phosphorylation levels of DARPP-32, which indicate its maximal activity, are reached in the media containing BSA, regardless of the presence of bicarbonate. Conversely, the lowest phosphorylation levels of DARPP-32 occurred in the medium without BSA and with 38 mM bicarbonate. These findings match with the results obtained from sperm function parameters, including membrane lipid disorder and acrosome exocytosis, and support that PKA activation, which is essential for sperm capacitation, can be achieved without adding bicarbonate to the capacitation medium. In contrast, BSA must be present, since the lack of that protein in the medium does not allow DARPP-32 to be phosphorylated.

The results obtained from the analysis of both MMP and O_2_ consumption rates suggest that mitochondria are activated at the start of the incubation in capacitation media, but only if BSA is present. This immediate effect on mitochondria in the presence of BSA takes place together with the activation of PKA and GSK3α. In this context, it is worth highlighting that phosphorylation of specific sperm proteins is crucial to reach the capacitated status [14]. In pigs, glycogen synthase kinase-3 (GSK3) is present in sperm cells and is phosphorylated at Ser9 and Ser21 in response to the activation of PKA, which completely activates GSK3 [75,76]. It has been reported that GSK3 activity is involved in the modulation of sperm motility, acrosome exocytosis and, ultimately, fertilization [75,77,78]. In this way, the results obtained from both PKA and GSK3α activities highlight how relevant BSA is for pig sperm to elicit in vitro capacitation and progesterone-induced acrosome exocytosis. In this respect, the mechanism through which BSA modulates the activities of PKA and GSK3α has been posited to be related to the activation of a CatSper-modulated extracellular Ca^2+^-influx [69,79]. Our results are in agreement with this mechanism, thus highlighting the relevance of BSA as an effector of capacitation in pig sperm. Furthermore, our data also suggest that serine as well astyrosine-phosphorylation regulate kinase activity of GSK3α and GSK3β in mammalian sperm [78,80,81,82,83]. Related to the specific role of serine residues, Rival et al. recently demonstrated that phosphatidylserine in mouse sperm plasmalemma is progressively exposed on the head region of viable spermatozoa during epididymal maturation [84]. Phosphatidylserine is recognized by specific oocyte membrane receptors, such as BAI1, CD36, TIM4, and MERTK, and when these phospholipids are masked, fertilization is inhibited.

Finally, it is worth mentioning that while this work aimed to address the effects of different concentrations of bicarbonate on the ability of pig sperm to elicit in vitro capacitation, a constant concentration of BSA was used. That concentration was chosen based on previous works, in which 5 mg/mL BSA was found to elicit in vitro capacitation of pig sperm [7,9,29,30]. Therefore, while this work specifically focused on the effects of bicarbonate, further studies should aim to question how different BSA concentrations modulate the response of pig sperm to the induction of in vitro capacitation and acrosome reaction. In addition, further research involving other indicators related to the underlying molecular mechanism, such as the phosphorylation of other sperm proteins and changes in intracellular cAMP levels, is likely to help gain new insights into how BSA and bicarbonate modulate PKA activation at different incubation times. On the other hand, although we did not find much variation between the pH of the different media throughout incubation (data not shown), we did not evaluate the potential impact that the different media composition could have on their osmolality. Hence, further studies investigating the effects of different BSA concentrations on how pig sperm elicit in vitro capacitation should also take this matter into consideration.

## 5. Conclusions

In conclusion, our results indicate that BSA is a crucial factor for pig sperm to elicit in vitro capacitation. In contrast, while the addition of bicarbonate to the medium is not a limiting factor to induce capacitation, the presence of high levels of bicarbonate reduces the time required for sperm to reach that status. In addition, based on all sperm parameters evaluated and on the phosphorylation levels of DARPP-32 and GSK3α, media containing BSA and either no or low levels of bicarbonate are the most suitable for inducing pig sperm capacitation. In effect, despite some capacitation events taking longer, sperm cells do not die as fast as the medium containing BSA and 38 mM bicarbonate when evaluated after 300 min of incubation.

## Figures and Tables

**Figure 1 biology-10-01105-f001:**
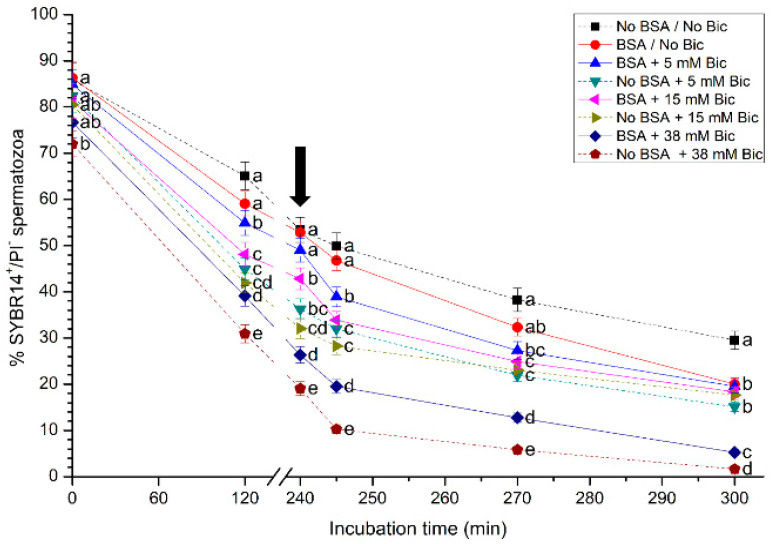
Percentages of sperm with an intact plasma membrane (SYBR14^+^/PI^−^) following incubation with different concentrations of bicarbonate (Bic; 0 mM, 5 mM, 15 mM, 38 mM) and BSA (0, 5 mg/mL) for 300 min. The black arrow indicates the time at which 10 μg/mL of progesterone was added (i.e., 240 min) to induce acrosome exocytosis. Different superscripts (a–e) indicate significant differences (*p* < 0.05) between treatments within the same time point. Data are shown as mean ± SEM for 12 independent experiments.

**Figure 2 biology-10-01105-f002:**
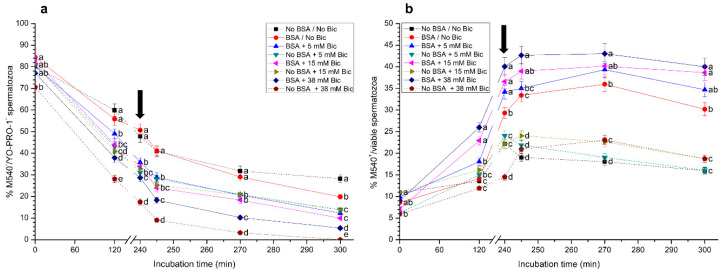
(**a**) Percentages of viable sperm with low membrane lipid disorder (M540^−^/YO-PRO-1^−^), and (**b**) percentages of sperm with high membrane lipid disorder considering the viable sperm population only (M540^+^/viable sperm) following incubation with different concentrations of bicarbonate (Bic; 0 mM, 5 mM, 15 mM, 38 mM) and BSA (0, 5 mg/mL) for 300 min. The black arrow indicates the time at which 10 μg/mL of progesterone was added (i.e., 240 min) to induce acrosome exocytosis. Different superscripts (a–e) indicate significant differences (*p* < 0.05) between treatments within the same time point. Data are shown as mean ± SEM for 12 independent experiments.

**Figure 3 biology-10-01105-f003:**
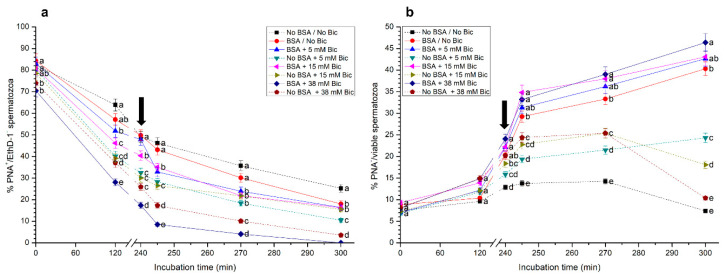
(**a**) Percentages of viable sperm with an intact acrosome (PNA^+^/EthD-1^−^), and (**b**) percentages of sperm with an exocytosed acrosome considering the viable sperm population only (PNA^−^/viable sperm) following incubation with different concentrations of bicarbonate (Bic; 0 mM, 5 mM, 15 mM, 38 mM) and BSA (0, 5 mg/mL) for 300 min. The black arrow indicates the time at which 10 μg/mL of progesterone was added (i.e., 240 min) to induce acrosome exocytosis. Different superscripts (a–e) indicate significant differences (*p*<0.05) between treatments within the same time point. Data are shown as mean ± SEM for 12 independent experiments.

**Figure 4 biology-10-01105-f004:**
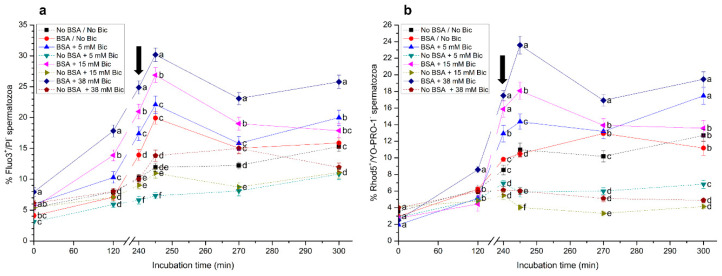
Percentages of viable sperm with high intracellular calcium levels evaluated through Fluo3^+^/PI^−^ (**a**) and Rhod5^+^/YO-PRO-1^−^ (**b**) following incubation with different concentrations of bicarbonate (Bic; 0 mM, 5 mM, 15 mM, 38 mM) and BSA (0, 5 mg/mL) for 300 min. The black arrow indicates the time at which 10 μg/mL of progesterone was added (i.e., 240 min) to induce acrosome exocytosis. Different superscripts (a–e) indicate significant differences (*p* < 0.05) between treatments within the same time point. Data are shown as mean ± SEM for 12 independent experiments.

**Figure 5 biology-10-01105-f005:**
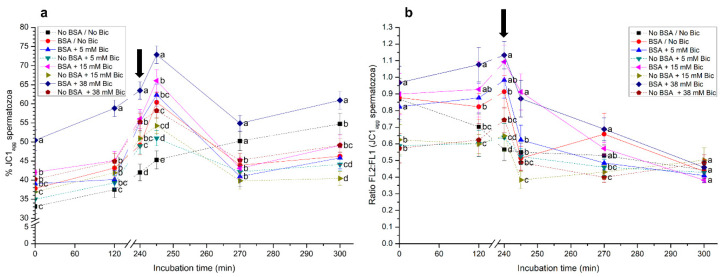
(**a**) Percentages of sperm with high mitochondrial membrane potential (MMP; JC1_agg_), and (**b**) FL2:FL1 (orange:green) ratios in the sperm population with high MMP (JC1_agg_) following incubation with different concentrations of bicarbonate (Bic; 0 mM, 5 mM, 15 mM, 38 mM) and BSA (0, 5 mg/mL) for 300 min. The black arrow indicates the time at which 10 μg/mL of progesterone was added (i.e., 240 min) to induce acrosome exocytosis. Different superscripts (a–d) indicate significant differences (*p* < 0.05) between treatments within the same time point. Data are shown as mean ± SEM for 12 independent experiments.

**Figure 6 biology-10-01105-f006:**
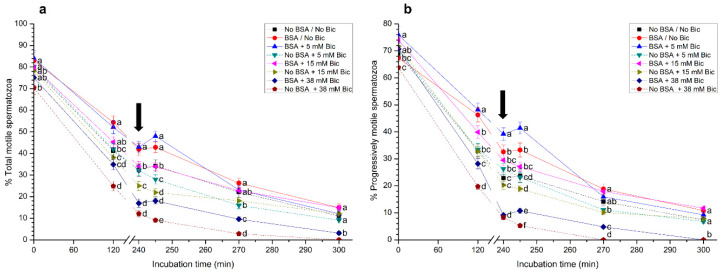
(**a**) Percentages of total and (**b**) progressively motile sperm following incubation with different concentrations of bicarbonate (Bic; 0 mM, 5 mM, 15 mM, 38 mM) and BSA (0, 5 mg/mL) for 300 min. The black arrow indicates the time at which 10 μg/mL of progesterone was added (i.e., 240 min) to induce acrosome exocytosis. Different superscripts (a–f) indicate significant differences (*p* < 0.05) between treatments within the same time point. Data are shown as mean ± SEM for 12 independent experiments.

**Figure 7 biology-10-01105-f007:**
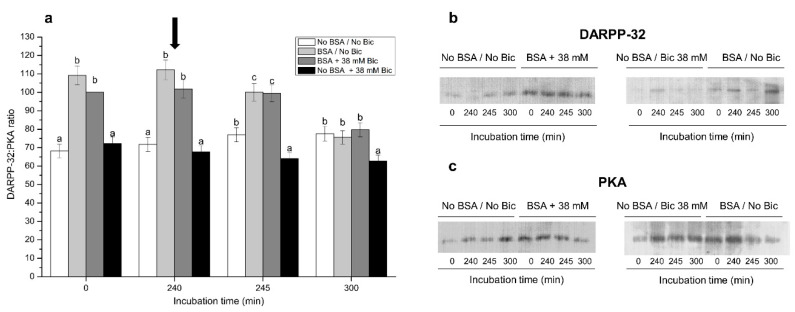
(**a**) Ratios between phosphorylated DARPP-32 and PKA following incubation with different concentrations of bicarbonate (Bic; 0 mM, 38 mM) and BSA (0, 5 mg/mL) for 300 min. The black arrow indicates the time at which 10 μg/mL of progesterone was added (i.e., 240 min) to induce acrosome exocytosis. Different superscripts (a–c) indicate significant differences (*p* < 0.05) between treatments within the same time point. Data were corrected to a basal arbitrary value of 100 for the control point, which corresponded to the incubation in the medium containing BSA and 38 mM bicarbonate at 0 min. Results are shown as mean ± SEM for 12 independent experiments. Representative blots for DARPP-32 (**b**) and PKA (**c**) after 0, 240, 245 and 300 min of incubation with No BSA/No Bic; BSA + 38 mM Bic; No BSA + 38 mM Bic; or BSA/No Bic.

**Figure 8 biology-10-01105-f008:**
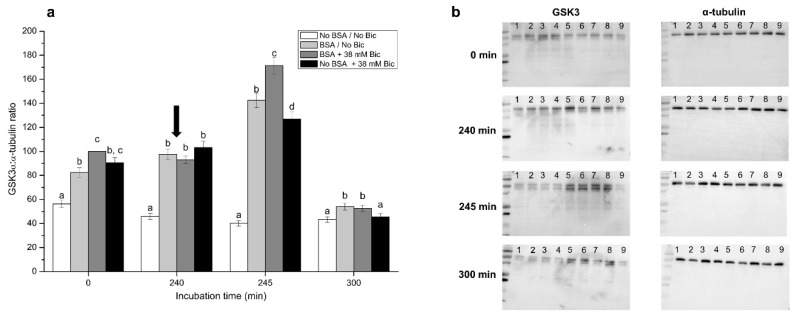
(**a**) Ratios between tyrosine-phosphorylated GSK3α and α-tubulin following incubation with different concentrations of bicarbonate (Bic; 0 mM, 38 mM) and BSA (0, 5 mg/mL) for 300 min. The black arrow indicates the time at which 10 μg/mL of progesterone was added (i.e., 240 min) to induce acrosome exocytosis. Different superscripts (a–d) indicate significant differences (*p* < 0.05) between treatments within the same time point. Data were corrected to a basal arbitrary value of 100 for the control point, which corresponded to the incubation in the medium containing BSA and 38 mM bicarbonate at 0 min. Results are shown as mean ± SEM for 12 independent experiments. (**b**) Representative blots for tyrosine-phosphorylated GSK3 and α-tubulin. Lane 1: No BSA/No Bic; Lane 2: BSA + 38 mM Bic; Lane 3: No BSA + 38 mM Bic; Lane 4: BSA/No Bic; Lane 5: No BSA + 5 mM Bic; Lane 6: BSA + 5 mM Bic; Lane 7: No BSA + 15 mM Bic; Lane 8: BSA + 15 mM Bic; Lane 9: No BSA/No Bic.

**Figure 9 biology-10-01105-f009:**
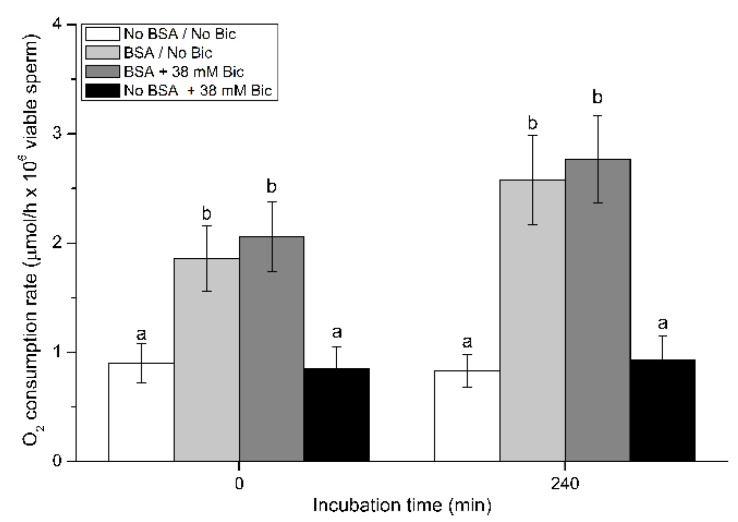
O_2_ consumption rate of pig sperm in the presence of different concentrations of bicarbonate (Bic; 0 mM, 38 mM) and BSA (0, 5 mg/mL) at 0 min and 240 min. Different superscripts (a,b) indicate significant differences (*p* < 0.05) between treatments within the same time point. Results are shown as mean ± SEM for 12 independent experiments.

**Table 1 biology-10-01105-t001:** Curvilinear velocity (VCL), straight-line velocity (VSL) and average path velocity (VAP) of pig sperm following incubation with different concentrations of bicarbonate (Bic; 0 mM, 5 mM, 15 mM, 38 mM) and BSA (0, 5 mg/mL) for 300 min. Progesterone (10 μg/mL) was added at 240 min to induce acrosome exocytosis. Different superscripts (a–d) indicate significant differences (*p* < 0.05) between treatments (rows) within the same time point. Data are shown as mean ± SEM for 12 independent experiments.

Media	0 Min	120 Min	240 Min	245 Min	270 Min	300 Min
**VCL (µm/s)**						
No BSA/No Bic	52.5 ± 3.1 ^a^	49.2 ± 2.8 ^a,b^	41.8 ± 2.5 ^a^	36.7 ± 2.1 ^a^	28.1 ± 1.6 ^a^	9.2 ± 0.6 ^a^
BSA/No Bic	63.4 ± 3.5 ^b^	56.1 ± 2.9 ^b,c^	54.3 ± 3.1 ^b^	51.2 ± 3.3 ^b^	49.5 ± 2.8 ^b^	47.8 ± 3.0 ^b^
BSA + 5 mM Bic	74.8 ± 3.7 ^c^	62.3 ± 3.5 ^c,d^	56.5 ± 3.1 ^b^	55.1 ± 3.0 ^b,c^	52.2 ± 3.2 ^b,c^	49.7 ± 2.8 ^b,c^
No BSA + 5 mM Bic	75.2 ± 4.2 ^c^	44.5 ± 2.8 ^a^	39.7 ± 2.4 ^a^	38.1 ± 2.1 ^a^	35.4 ± 2.3 ^a^	10.3 ± 0.4 ^a^
BSA + 15 mM Bic	76.2 ± 4.1 ^c^	63.7 ± 3.8 ^c,d^	58.5 ± 3.3 ^b^	57.8 ± 3.1 ^b,c^	54.0 ±3.3 ^b,c^	51.3 ± 2.9 ^b,c^
No BSA + 15 mM Bic	70.8 ± 4.6 ^b,c^	52.3 ± 2.9 ^a,b^	41.2 ± 2.2 ^a^	39.8 ± 2.3 ^a^	32.3 ±1.8 ^a^	8.9 ± 0.4 ^a^
BSA + 38 mM Bic	75.9 ± 4.3 ^c^	66.3 ± 4.0 ^d^	60.5 ± 3.8 ^b^	62.1 ± 3.5 ^c^	59.3 ±3.4 ^c^	55.6 ± 3.8 ^c^
No BSA + 38 mM Bic	67.4 ± 3.8 ^b,c^	54.2 ± 3.1 ^b^	34.8 ± 2.1 ^a^	32.2 ± 1.9 ^a^	19.1 ±1.1 ^d^	3.0 ± 0.2 ^c^
**VSL (µm/s)**						
No BSA/No Bic	25.4 ± 1.6 ^a^	19.2 ± 1.1 ^a^	13.4 ± 0.9 ^a^	10.3 ± 0.6 ^a^	6.7 ± 0.4 ^a^	3.8 ± 0.1 ^a^
BSA/No Bic	33.8 ± 2.0 ^a^	27.5 ± 1.8 ^b^	25.6 ± 1.5 ^b^	25.1 ± 1.5 ^b^	23.2 ± 1.4 ^b^	20.9 ± 1.3 ^b^
BSA + 5 mM Bic	32.5 ± 2.1 ^a^	29.8 ± 1.9 ^b^	28.3 ± 1.7 ^b^	26.4 ± 1.4 ^b^	22.3 ± 1.3 ^b^	19.5 ± 1.2 ^b^
No BSA + 5 mM Bic	30.1 ± 1.8 ^a^	23.2 ± 1.4 ^a,b^	18.4 ± 1.2 ^a^	15.6 ± 0.9 ^a^	9.6 ± 0.5 ^a^	3.2 ± 0.2 ^a^
BSA + 15 mM Bic	31.4 ± 2.1 ^a^	30.3 ± 1.9 ^b^	26.2 ± 1.7 ^b^	25.8 ± 1.8 ^b^	27.1 ± 1.5 ^b^	22.9 ± 1.3 ^b^
No BSA + 15 mM Bic	30.5 ± 2.2 ^a^	18.3 ± 1.2 ^a^	17.1 ± 1.0 ^a^	12.8 ± 0.8 ^a^	7.2 ± 0.5 ^a^	1.8 ± 0.1 ^a^
BSA + 38 mM Bic	33.5 ± 2.1 ^a^	31.8 ± 2.3 ^b^	25.6 ± 1.7 ^b^	23.4 ± 1.4 ^b^	20.2 ± 1.3 ^b^	18.3 ± 1.0 ^b^
No BSA + 38 mM Bic	29.2 ± 2.0 ^a^	26.7 ± 1.8 ^a,b^	15.4 ± 1.1 ^a^	11.3 ± 0.7 ^a^	9.6 ± 0.6 ^a^	1.4 ± 0.1 ^a^
**VAP (µm/s)**						
No BSA/No Bic	40.3 ± 2.5 ^a^	34.8 ± 2.2 ^a^	26.2 ± 1.8 ^a^	27.1 ± 1.7 ^a^	19.2 ± 1.2 ^a^	5.6 ± 0.4 ^a^
BSA/No Bic	42.8 ± 2.8 ^a,b^	41.3 ± 2.7 ^a,b^	38.9 ± 2.5 ^b,c^	40.1 ± 2.8 ^b^	39.5 ± 2.5 ^b^	37.0 ± 2.6 ^b^
BSA + 5 mM Bic	46.1 ± 2.6 ^a,b^	43.5 ± 2.4 ^b^	40.6 ± 2.2 ^c^	48.2 ± 2.7 ^b^	42.5 ± 2.3 ^b^	43.4 ± 2.4 ^b^
No BSA + 5 mM Bic	44.1 ± 2.5 ^a,b^	39.2 ± 2.3 ^a,b^	30.1 ± 2.1 ^a^	21.4 ± 1.3 ^a^	18.5 ± 1.1 ^a^	7.6 ± 0.5 ^a^
BSA + 15 mM Bic	46.7 ± 2.8 ^a,b^	40.1 ± 2.5 ^a,b^	38.4 ± 2.3 ^b,c^	43.2 ± 2.6 ^b^	37.0 ± 2.2 ^b^	38.1 ± 2.0 ^b^
No BSA + 15 mM Bic	42.3 ± 2.6 ^a,b^	38.1 ± 2.4 ^a,b^	32.5 ± 2.2 ^a,b^	23.8 ± 1.5 ^a^	20.7 ± 1.3 ^a^	5.5 ± 0.3 ^a^
BSA + 38 mM Bic	49.5 ± 2.8 ^b^	43.3 ± 2.6 ^b^	42.8 ± 2.6 ^c^	48.7 ± 3.0 ^b^	45.2 ± 2.5 ^b^	35.8 ± 2.2 ^b^
No BSA + 38 mM Bic	44.6 ± 2.4 ^a,b^	40.8 ± 2.3 ^a,b^	29.1 ± 1.8 ^a^	22.4 ± 1.4 ^a^	15.6 ± 1.0 ^a^	2.2 ± 0.2 ^a^

**Table 2 biology-10-01105-t002:** Percentages of linearity (LIN), straightness (STR) and oscillation (WOB) following incubation of pig sperm with different concentrations of bicarbonate (Bic; 0 mM, 5 mM, 15 mM, 38 mM) and BSA (0, 5 mg/mL) for 300 min. Progesterone (10 μg/mL) was added at 240 min to induce acrosome exocytosis. Different superscripts (a–e) indicate significant differences (*p* < 0.05) between treatments (rows) within the same time point. Data are shown as mean ± SEM for 12 independent experiments.

Media	0 Min	120 Min	240 Min	245 Min	270 Min	300 Min
**LIN (%)**						
No BSA/No Bic	48.4 ± 2.9 ^a,b^	39.0 ± 2.3 ^a,c^	32.1 ± 1.8 ^a^	28.1 ± 1.6 ^a^	23.8 ± 1.4 ^a^	41.3 ± 2.5 ^a^
BSA/No Bic	53.3 ± 3.1 ^a^	49.0 ± 2.8 ^b^	47.1 ± 2.7 ^b^	49.0 ± 2.8 ^b^	46.9 ± 2.7 ^b^	43.7 ± 2.6 ^a^
BSA + 5 mM Bic	43.4 ± 2.5 ^b^	47.8 ± 2.8 ^a,b^	50.1 ± 2.9 ^b^	47.9 ± 2.8 ^b^	42.7 ± 2.4 ^b,d^	39.2 ± 2.3 ^a,b^
No BSA + 5 mM Bic	40.0 ± 2.4 ^b^	52.1 ± 3.0 ^b^	46.3 ± 2.7 ^b^	40.9 ± 2.4 ^b,c^	27.1 ±1.6 ^a,c^	31.1 ± 1.8 ^b^
BSA + 15 mM Bic	41.2 ± 2.3 ^b^	47.6 ± 2.8 ^a,b^	44.8 ± 2.6 ^b^	44.6 ± 2.5 ^b,c^	50.2 ± 2.9 ^b^	44.6 ± 2.6 ^a^
No BSA + 15 mM Bic	43.1 ± 2.5 ^b^	35.0 ± 2.0 ^c^	41.5 ± 2.4 ^b^	32.2 ± 1.9 ^a^	22.3 ± 1.3 ^a^	20.2 ± 1.2 ^c^
BSA + 38 mM Bic	44. 1± 2.6 ^b^	48.0 ± 2.8 ^a,b^	42.3 ± 2.5 ^b^	37.7 ± 2.2 ^a,c^	34.1 ± 2.0 ^c,d^	32.9 ± 1.9 ^b^
No BSA + 38 mM Bic	43.3 ± 2.6 ^b^	49.3 ± 2.9 ^b^	44.3 ± 2.7 ^b^	35.1 ± 2.1 ^a,c^	50.3 ± 3.0 ^b^	46.7 ± 2.8 ^a^
**STR (%)**						
No BSA/No Bic	63.0 ± 3.5 ^a^	55.2 ± 3.2 ^a,b^	51.1 ± 3.0 ^a^	38.0 ± 2.3 ^a^	34.9 ± 2.1 ^a^	67.9 ± 3.6 ^a^
BSA/No Bic	79.0 ± 4.4 ^b^	66.6 ± 3.8 ^c,d,e^	65.8 ± 3.7 ^b,c^	62.6 ± 3.6 ^b^	58.7 ± 3.5 ^b^	56.5 ± 3.2 ^b,e^
BSA + 5 mM Bic	70.5 ± 4.1 ^a,b^	68.5 ± 3.9 ^c,d,e^	69.7 ± 4.0 ^c^	54.8 ± 3.1 ^b^	52.5 ± 3.1 ^b,c^	44.9 ± 2.6 ^c^
No BSA + 5 mM Bic	68.3 ± 3.8 ^a^	59.2 ± 3.4 ^b,e^	61.1 ± 3.5 ^b,c^	72.9 ± 4.1 ^c^	51.9 ± 2.9 ^b,c^	42.1 ± 2.4 ^c^
BSA + 15 mM Bic	67.2 ± 3.8 ^a^	75.6 ± 4.2 ^c^	68.2 ± 3.8 ^b,c^	59.7 ± 3.4 ^b^	73.2 ± 4.1 ^d^	60.1 ± 3.5 ^a,b,e^
No BSA + 15 mM Bic	72.1 ± 4.2 ^a,b^	48.0 ± 2.8 ^a^	52.6 ± 3.1 ^a^	53.8 ± 3.0 ^b^	34.8 ± 2.0 ^a^	32.7 ± 1.9 ^d^
BSA + 38 mM Bic	67.7 ± 3.9 ^a^	73.4 ± 4.1 ^c,d^	59.8 ± 3.4 ^a,b^	48.0 ± 2.8 ^a,b^	44.7 ± 2.6 ^c^	51.1 ± 2.9 ^b,c^
No BSA + 38 mM Bic	65.5 ± 3.8 ^a^	65.4 ± 3.3 ^b,d,e^	52.9 ± 3.0 ^a^	50.4 ± 2.9 ^a,b^	61.5 ± 3.5 ^b^	63.6 ± 3.7 ^a,e^
**WOB (%)**						
No BSA/No Bic	76.8 ± 4.2 ^a^	70.7 ± 4.1 ^a,b^	62.7 ± 3.5 ^a^	73.8 ± 4.3 ^a^	68.3 ± 3.8 ^a,b^	60.9 ± 3.4 ^a^
BSA/No Bic	67.5 ± 3.9 ^a^	73.6 ± 4.1 ^a^	71.6 ± 4.2 ^a,b^	78.3 ± 4.3 ^a,b^	79.8 ± 4.5 ^d^	77.4 ± 4.4 ^b^
BSA + 5 mM Bic	61.6 ± 3.4 ^b^	69.8 ± 3.9 ^a,b^	71.9 ± 4.1 ^a,b^	87.5 ± 4.8 ^b^	81.4 ± 4.7 ^d^	87.3 ± 4.8 ^c^
No BSA + 5 mM Bic	58.6 ± 3.3 ^b^	88.1 ± 4.8 ^c^	75.8 ± 4.2 ^b,c^	56.2 ± 3.1 ^c^	52.3 ± 2.9 ^c^	73.8 ± 4.2 ^b^
BSA + 15 mM Bic	61.3 ± 3.3 ^b^	63.0 ± 3.6 ^b^	65.6 ± 3.7 ^a,b^	74.7 ± 4.2 ^a^	68.5 ± 3.8 ^a,b^	74.3 ± 4.1 ^b^
No BSA + 15 mM Bic	59.7 ± 3.4 ^b^	72.8 ± 4.1 ^a,b^	78.9 ± 4.4 ^c^	59.8 ± 3.4 ^c^	64.1 ± 3.7 ^a^	61.8 ± 3.6 ^a^
BSA + 38 mM Bic	65.2 ± 3.7 ^a,b^	65.3 ± 3.5 ^b^	70.7 ± 4.2 ^a,b^	78.4 ± 4.5 ^a,b^	76.2 ± 4.3 ^b,d^	64.4 ± 3.7 ^a^
No BSA + 38 mM Bic	66.2 ± 3.7 ^a,b^	75.3 ± 4.2 ^a^	83.6 ± 4.7 ^c^	69.6 ± 4.0 ^a^	81.7 ± 4.6 ^d^	73.3 ± 4.1 ^b^

**Table 3 biology-10-01105-t003:** Amplitude of lateral head displacement (ALH) and frequency of head displacement (BCF) following incubation of pig sperm with different concentrations of bicarbonate (Bic; 0 mM, 5 mM, 15 mM, 38 mM) and BSA (0, 5 mg/mL) for 300 min. Progesterone (10 μg/mL) was added at 240 min to induce acrosome exocytosis. Different superscripts (a–d) indicate significant differences (*p* < 0.05) between treatments (rows) within the same time point. Data are shown as mean ± SEM for 12 independent experiments.

Media	0 Min	120 Min	240 Min	245 Min	270 Min	300 Min
**ALH (µm)**						
No BSA/No Bic	2.5 ± 0.1 ^a^	2.4 ± 0.1 ^a^	2.6 ± 0.1 ^a^	2.1 ± 0.1 ^a^	2.2 ± 0.1 ^a^	1.9 ± 0.1 ^a^
BSA/No Bic	2.7 ± 0.1 ^a^	3.2 ± 0.2 ^b^	3.7 ± 0.2 ^b^	3.9 ± 0.2 ^b^	4.1 ± 0.2 ^b^	3.8 ± 0.2 ^b^
BSA + 5 mM Bic	2.6 ± 0.1 ^a^	3.1 ± 0.2 ^b^	3.9 ± 0.2 ^b^	4.1 ± 0.2 ^b^	4.3 ± 0.2 ^b^	4.2 ± 0.2 ^b^
No BSA + 5 mM Bic	2.3 ± 0.1 ^a^	2.2 ± 0.1 ^a^	2.4 ± 0.1 ^a^	2.3 ± 0.1 ^a^	1.9 ± 0.1 ^a^	1.7 ± 0.1 ^a^
BSA + 15 mM Bic	2.5 ± 0.1 ^a^	3.3 ± 0.2 ^b^	3.4 ± 0.2 ^b^	3.6 ± 0.2 ^b^	4.0 ± 0.2 ^b^	3.9 ± 0.2 ^b^
No BSA + 15 mM Bic	2.2 ± 0.1 ^a^	2.4 ± 0.1 ^a^	2.6 ± 0.1 ^a^	2.5 ± 0.1 ^a^	2.3 ± 0.1 ^a^	2.0 ± 0.1 ^a^
BSA + 38 mM Bic	2.6 ± 0.1 ^a^	3.5 ± 0.2 ^b^	3.7 ± 0.2 ^b^	3.8 ± 0.2 ^b^	3.6 ± 0.2 ^b^	3.5 ± 0.2 ^b^
No BSA + 38 mM Bic	2.1 ± 0.1 ^a^	2.4 ± 0.1 ^a^	2.5 ± 0.1 ^a^	2.3 ± 0.1 ^a^	2.1 ± 0.1 ^a^	1.8 ± 0.1 ^a^
**BCF (Hz)**						
No BSA/No Bic	6.8 ± 0.4 ^a,b^	5.1 ± 0.3 ^a^	4.9 ± 0.3 ^a^	5.2 ± 0.2 ^a^	4.7 ± 0.2 ^a^	3.8 ± 0.2 ^a^
BSA/No Bic	6.5 ± 0.3 ^a^	6.5 ± 0.3 ^b^	7.3 ± 0.4 ^b^	7.1 ± 0.4 ^b^	6.7 ± 0.3 ^b^	6.4 ± 0.3 ^b^
BSA + 5 mM Bic	6.1 ± 0.3 ^a^	6.7 ± 0.3 ^b,c^	7.6 ± 0.4 ^b^	7.2 ± 0.4 ^b^	6.7 ± 0.3 ^b^	6.5 ± 0.3 ^b^
No BSA + 5 mM Bic	6.2 ± 0.3 ^a^	5.9 ± 0.3 ^a,b^	5.5 ± 0.3 ^a,d^	5.3 ± 0.3 ^a^	5.1 ± 0.3 ^a^	4.9 ± 0.2 ^c^
BSA + 15 mM Bic	7.2 ± 0.4 ^b^	7.1 ± 0.4 ^c,d^	6.4 ± 0.3 ^c^	6.9 ± 0.3 ^b^	6.5 ± 0.3 ^b^	6.2 ± 0.3 ^b^
No BSA + 15 mM Bic	6.8 ± 0.4 ^a,b^	6.2 ± 0.3 ^b,c^	5.8 ± 0.3 ^c,d^	5.5 ± 0.3 ^a^	5.4 ± 0.3 ^a^	5.1 ± 0.3 ^c^
BSA + 38 mM Bic	7.4 ± 0.4 ^b^	7.7 ± 0.4 ^d^	7.5 ± 0.4 ^b^	7.2 ± 0.4 ^b^	6.3 ± 0.3 ^b^	6.0 ± 0.3 ^b^
No BSA + 38 mM Bic	7.1 ± 0.4 ^b^	6.4 ± 0.3 ^b,c^	5.9 ± 0.3 ^c,d^	5.6 ± 0.3 ^a^	5.3 ± 0.3 ^a^	5.2 ± 0.3 ^c^

## Data Availability

The data presented in this study are available in the article and Appendix A.

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
