# Peer review of "Exogenous Albumin Is Crucial for Pig Sperm to Elicit In Vitro Capacitation Whereas Bicarbonate Only Modulates Its Efficiency"

_biology, 2021, doi:10.3390/biology10111105_

Round 1
Reviewer 1 Report
The authors answered all of my questions. I accept the work in the present form
Author Response
General Comment: “The authors answered all of my questions. I accept the work in the present form”.
Answer: We really appreciate your positive assessment and we would like to thank you for kindly revising our Manuscript and give us this feedback.

Reviewer 2 Report
It is a paper with great scientific and laboratory work.
Perhaps in its structure, the chapter on material and methods, and the results chapter is excessively long, with respect to the discussion and conclusions.
In the CASA parameters in the results chapter, those that do not contribute information to the discussion or come from the formulas of the others could be eliminated. For example LIN, STR, WOB. It is enough to comment on the significant details.
Author Response
Reviewer #2
General Comments
General Comment: “It is a paper with great scientific and laboratory work. Perhaps in its structure, the chapter on material and methods, and the results chapter is excessively long, with respect to the discussion and conclusions. In the CASA parameters in the results chapter, those that do not contribute information to the discussion or come from the formulas of the others could be eliminated. For example LIN, STR, WOB. It is enough to comment on the significant details.”
Answer: Thank you very much for your comment and consideration. We have revised the results to avoid an excessive detailed comment on some CASA parameters. That being said, in previous revisions, we were requested by other reviewers to provide these data.